# Physicochemical Characteristics and Microstructure of Ancient and Common Wheat Grains Cultivated in Romania

**DOI:** 10.3390/plants12112138

**Published:** 2023-05-29

**Authors:** Camelia Maria Golea, Silviu-Gabriel Stroe, Anca-Mihaela Gâtlan, Georgiana Gabriela Codină

**Affiliations:** 1Faculty of Food Engineering, “Ştefan cel Mare” University, 720229 Suceava, Romania; 2Vegetal Genetic Resources Bank “Mihai Cristea”, 720224 Suceava, Romania

**Keywords:** wheat, proximate composition, minerals, grain microstructure, principal component analysis

## Abstract

Different wheat species, common wheat (*Triticum aestivum* subsp. aestivum), spelt (*Triticum aestivum* subsp. *spelta*) and einkorn (*Triticum monococcum* subsp. *monococcum*), were analyzed for physicochemical (moisture, ash, protein, wet gluten, lipid, starch, carbohydrates, test weight and thousand-kernel mass) and mineral elements (Ca, Mg, K, Na, Zn, Fe, Mn and Cu) concentrations in grains. Additionally, wheat grain microstructure was determined using a scanning electron microscope. SEM micrographs of wheat grains show that einkorn has smaller type A starch granule diameters and more compact protein bonds compared to common wheat and spelt grains, making it easier to digest. The ancient wheat grains presented higher values for ash, protein, wet gluten and lipid content compared to the common wheat grains, whereas the carbohydrates and starch content were significantly (*p* < 0.05) lower. The mean values showed that spelt (*Triticum aestivum* subsp. *spelta*) grains presented the highest values for Ca, Mg and K, while einkorn (*Triticum monococcum* subsp. *monococcum*) grains had the highest values for the microelements Zn, Mn and Cu. The highest values of Fe were recorded for common wheat varieties whereas no significant differences among the species were obtained for Na content. The principal component analysis (*p* > 0.05) between wheat flours characteristics showed a close association between wheat grain species and between the chemical characteristics of gluten and protein content (r = 0.994), lipid and ash content (r = 0.952) and starch and carbohydrate content (r = 0.927), for which high positive significant correlations (*p* < 0.05) were obtained. Taking into account that Romania is the fourth largest wheat producer at the European level, this study is of great global importance. According to the results obtained, the ancient species have higher nutritional value from the point of view of chemical compounds and macro elements of minerals. This may be of great importance for consumers who demand bakery products with high nutritional quality.

## 1. Introduction

Cereals are the most important food products for humans. From these, common wheat (*Triticum aestivum* subsp. *aestivum*), together with rye, are the only grains used for baking that are cultivated in large areas, their production being in continuous development. The largest global wheat producers in the 2020–2021 season were China, India, Russia, the USA, Canada, Ukraine, Pakistan and the EU [1]. In the EU, Romania is one of the most important wheat producers, ranked fourth highest after France, Germany and Poland [2]. From a nutritional and economic point of view, no food satisfies human requirements as wheat bread does. Wheat is rich in proteins, which are represented by prolamins, glutenins, globulins and albumins [3]. They ensure the growth and development of the body and have a very important biocatalytic and energetic role. Wheat contains a large amount of starch, the main component of the grain, as well as some fermentable sugars (maltose and sucrose) [4]. All these have a very important energetic role. Wheat germ contains 2–4% lipids that are rich in unsaturated fatty acids, as well as vitamin E [5]. Wheat is rich in vitamins from group B (B1, B2, B5, B6), vitamin PP and K (K1, K2, K3) [6]. From the point of view of mineral substances, wheat is rich in P, K and Mg, and also may contain significant amounts of Fe [7].

Among the many species of wheat that are cultivated, the most common is common wheat (*Triticum aestivum* subsp. *aestivum*), which occupies about 95% of the area cultivated with wheat throughout the globe [8]. This fact is due to the high-yielding productivity of this species, but also because of its special characteristics for bread-making [9]. Ancient wheat species: spelt (*Triticum aestivum* subsp. *spelta*), einkorn (*Triticum monococcum,* subsp. *monococcum)* and emmer (*Triticum turgidum* subsp. *dicoccum*) were the primary food crop in the human diet during the Bronze and Neolithic ages. Ancient wheat’s cultivation is not as extensive due to its hulled kernel and its low productivity [10] and, since 1960, has been drastically replaced with modern varieties [9]. However, nowadays consumers’ interest in food products with high nutritional value has led to a rediscovery of ancient wheat species and their cultivation has begun to expand. Moreover, they are superior to common wheat because of their tolerance of biotic and abiotic stresses such as pests, diseases, drought, cold, heat, pollution, salinity and soil nutrient shortage [6]. According to different studies, the nutritional value of ancient grains is higher than those of modern ones [11,12]. Thus, they have a higher level of lipids (mostly unsaturated fatty acids), proteins, soluble fibers, vitamins, minerals and phytochemicals [11]. Einkorn (*Triticum monococcum* subsp. *monococcum)* and spelt (*Triticum aestivum* subsp. *spelta*) contain more fat-soluble vitamins (D, A, E and B) and more easily absorbed protein and microelements (Fe, P, Zn, Ca and Mn) [13]. Wheat is an important source of minerals for the human body. For example, bread provides 15% of the daily intake of Fe, 13% of Mg, 11% of Zn and 14% of Cu [14]. The mineral content of wheat products varies depending on the species and the degree of extraction of the flour. Einkorn (*Triticum monococcum* subsp. *monococcum)* grains species have a smaller size, which increases the amount of bran to flour, leading to a concentration effect of mineral elements presented in higher levels in the bran fraction [15]. Minerals are essential for humans, stimulating developmental functions such as growth, energy production, well-balanced blood, healing and bone development, muscle regulation and the maintenance of a healthy nervous system, and are part of many enzymes [16]. The mineral amount in wheat grains may be environmentally determined or may be due to the cultivar choice. Ancient wheat grains such as einkorn (*Triticum monococcum* subsp. *monococcum)* and spelt (*Triticum aestivum* subsp. *spelta*) have been found to have high mineral content of P, Mg, Zn and Se compared to common wheat (*Triticum aestivum* subsp. *aestivum*) grains [17]. Different authors have also reported higher amounts of Fe, Mn, Zn and Cu in einkorn (*Triticum monococcum* subsp. *monococcum*), detecting a large genotypic variation [18]. Although much information on ancient and modern wheat grains has been published previously, very few data are available on wheat grains cultivated in Romania, one of the most important wheat producers in the EU and therefore the world. Within this study, we analyze the physicochemical and mineral elements (Ca, Mg, K, Na, Zn, Fe, Mn and Cu) using flame atomic absorption spectrometry of different wheat varieties cultivated in Romania in the same conditions. These mineral elements were in the highest amount in these wheat grains as we have previously reported in our study using an EDX system (Energy Dispersive X-ray Analysis), a less precise method which only quantifies these mineral elements, for their evaluation [19]. Additionally, we analyze the wheat grains’ microstructure using a scanning electron microscope. To our knowledge, this is the first study that conducts a complete overview on wheat grains’ physicochemical composition, including mineral elements evaluation and their microstructure analysis for different wheat varieties cultivated in Romania.

## 2. Results

### 2.1. Wheat Flour Physicochemical Characteristics

The wheat flour physiochemical characteristics are shown in Table 1. According to the data obtained, there are significant differences (*p* < 0.001) between wheat grains for all wheat chemical characteristics. The ancient wheat grains presented higher values for ash, protein, wet gluten and lipid content compared to the common wheat grains. This confirms the results of other studies, which have also reported that ancient wheat grains are richer in these compounds than common wheat [13,20,21]. According to Biel et al. [13], the highest protein, lipid and ash content were found in einkorn and the lowest in common wheat. According to Golea et al. [20], the highest protein and wet gluten content were obtained for einkorn, followed by spelt and common wheat, whereas the highest fat and ash content were obtained for spelt, followed by einkorn and common wheat grains. Athinaiou et al. [21] reported that ancient wheat species presented higher levels of protein, fat and ash than common wheat grains. From these, the highest values for ash and lipid content were obtained for spelt grains. Carbohydrate and starch content of ancient wheat grains were significantly lower (*p* < 0.001) than in modern grains, which is in agreement with the literature reports [22]. This is due to the fact that these grains accumulate higher amounts of proteins, which will lead to lower starch content. The variations in carbohydrates are according to expectations, since the rest of the grain compounds, namely fat, ash and protein, are higher in einkorn and spelt grains than in common wheat flours. Thousand-kernel mass (TKM) of spelt was higher than that of common wheat. In contrast, the mean value of einkorn was lower compared to common wheat, which is in agreement with data reported by others [23,24]. The test weight values did not present significant differences (*p* > 0.05) among the wheat samples. The highest grain test weight mean value was recorded for common wheat grains, probably due to their larger size compared to ancient wheat grains.

### 2.2. Wheat Flour Mineral Elements

The analyzed values of the wheat flour mineral elements are shown in Table 2. Among the analyzed minerals, K had the highest concentration, followed by Mg, Ca and Na. This order was found for each analyzed wheat species. A similar descending order of mineral elements in wheat grains was reported by Biel et al. [13]. According to them, for all ancient and common wheat species, the following descending order was also found for the analyzed mineral macro elements: K ˃ Mg ˃ Ca. The samples of spelt species had higher mean Ca, Mg, K and Na concentration. No significant differences were obtained for Na (*p* > 0.05), for which common wheat presented the highest mean value. Except for Fe, all the microelements analyzed presented the highest values in einkorn grains. These data are in agreement with those obtained by Tekin et al. [25], who reported positive correlations for einkorn grains and Zn, Mn and Cu.

Unexpectedly, Fe concentrations were higher in common wheat compared to ancient varieties, which is in disagreement with the literature reports [6]. However, Zhao et al. [26] also reported that spelt and einkorn did not contain high levels of Fe. The concentration of Cu presented significantly higher values (*p* < 0.001) for ancient grains compared to the modern varieties, these data being in agreement with those reported by others [6].

### 2.3. Wheat Grain Microstructure

We performed an imaging study based on a scanning electron microscopy (SEM) analysis in order to describe the main structural layers of the wheat caryopsis. Therefore, Figure 1 shows the microstructures of mature grains of common wheat (A) and einkorn (B) species, cross-sectioned.

According to the wheat caryopsis micrographs shown in Figure 1, three important and morphologically different tissues were observed, as follows: the pericarp layer (P) fused with the seed coat (SC), the aleurone layer (A) and the starchy endosperm layer (E). The fourth main tissue, an important component of the wheat caryopsis, is the embryo, but it cannot be seen in these micrographs, as it is located on the dorsal side of the grain. The SEM micrographs were evaluated using ImageJ software (National Institutes of Health, USA), the measurements being made after an appropriate calibration. Thus, the first layer, represented by pericarp (P), is a lignified, dead tissue that presents different thicknesses in both cases: 15 µm in common wheat and 8 µm in einkorn. On the other hand, the seed coat, which also showed a distinct thickness for each species: 11 µm for common wheat and 5 µm for einkorn, represents the outer layer that completely covers the seed and fuses with the pericarp (P).

A thick layer of cells, which represents the outer layer of the endosperm, called the aleurone layer (A), which can be seen in Figure 2, was captured during einkorn SEM analysis.

The aleurone layer (A) has visibly different thicknesses in both species: 43 µm in common wheat and 25 µm in einkorn. Aleurone cells represent living tissue at maturity, and are filled with protein bodies or globoids (G) and lipid droplets. Globoids were observed as white beads [27] with a diameter between 1.06–2.76 µm in the case of common wheat and 1.8–4.8 µm in the case of einkorn, respectively. According to Panato et al. [28], the structure and properties of these granules are influenced by the amount of lipids associated with the starch. Additionally, the empty space that appears between the walls and aleurone cells is due to dehydration and handling of the sample during section preparation [27].

Figure 3 shows SEM images of the starch endosperm of 12 wheat samples, which belong to 3 distinct species: common wheat (A), einkorn (B) and spelt (C).

SEM micrographs show that type A and B granules, distributed throughout the protein matrix, are evident and can be separated visually in all three analyzed species. Therefore, the diameters of the starch granules, determined using ImageJ software, had lower values than those found in the literature, as follows: type A granules—16.63 ± 3.53 µm in common wheat, 12.59 ± 4.99 µm in einkorn and 16.39 ± 5.4 µm in spelt, respectively; and type B granules—4.38 ± 2.25 µm in common wheat, 3.85 ± 1.69 µm in einkorn and 3.23 ± 1.45 µm in spelt, respectively.

### 2.4. Principal Component Analysis between Wheat Flours Characteristics

PCA loadings of the wheat flours characteristics are represented in Figure 4. The two plots, PC1 and PC2, represent 49.41% and 17.48% of the total variance. The PCA graph shows, along the PC1 and PC2 axes, a close association between the chemical characteristics of gluten and protein content (r = 0.994), lipid and ash content (r = 0.952) and starch and carbohydrate content (r = 0.927), for which high positive significant correlations (*p* < 0.05) were obtained. Similar positive correlations between these chemical characteristics have also been reported by others [20,29,30,31]. All the wheat grain species are closely associated on the PCA graph. Along the PC2 axis, the ancient wheat grains were positioned on the PCA graph on the right, whereas the modern wheat grain varieties were positioned on the left. Additionally, the PC2 axes clearly distinguish the ancient species, einkorn being placed on the bottom right and spelt placed on the upper right part of the PCA graph. A close association between different wheat species has also been reported by others [20,29]. Additionally, the second principal component PC2 shows a close association between ancient wheat flour species and their chemical characteristics of ash, lipids, gluten, proteins, Mn, Zn, K, Cu and Ca, and between modern wheat flours and their physicochemical characteristics of carbohydrates, starch, moisture, hectolitric weight, thousand-kernel mass, Fe, Na and Mg. However, PC1 shows a closer association between einkorn and the chemical characteristics of lipids, ash, Zn and Mn, and between the spelt and chemical characteristic of gluten, protein, Ca, Cu and K.

## 3. Discussion

The moisture content of the wheat grains was less than 14% for all the analyzed samples, meaning that all wheat grains had a long shelf life during storage [20]. However, ancient grains presented significantly lower (*p* < 0.05) moisture values than modern grains, probably due to the fact that these grains are hulled, which protects the kernel from the outside humidity. The highest ash values were noted in einkorn grains, followed by spelt and common wheat. This may be due to the higher share of seed coat in the ancient kernels compared to the modern kernels [32]. Additionally, significantly higher values (*p* < 0.05) were recorded for lipid content from ancient grains compared to the modern grains. This fact was explainable by both lipids and mineral content (represented by ash value) being located in the outer layers and germ of the wheat kernel [20]. The protein content of ancient wheat species was significantly higher (*p* < 0.05) than those of modern grains. This shows that, in spite of the high level of cultivation technology, the ancient wheat species can retain higher amounts of proteins compared to the modern wheat species. However, this high amount of protein in hulled wheat grains may be a consequence of low grain yield. Weight and grain size are important characteristics in wheat chemical compounds, such that a high size and heavy grain, for example, yield a smaller proportion of external pericarp and aleuronic layers, as well as larger endosperm. A wheat grain is formed from three main components: outer layers (aleurone, seed coat and pericarp), endosperm and germ (embryo). Most of the dry weight of the wheat (approximately 90%) is accounted for by endosperm. According to Azrani and Ashraf [6], the lower protein content of modern wheat may be due to its heavier and larger grain that yields a larger starchy endosperm, which, in turn, lessens its protein content [6]. The wet gluten is strongly correlated with protein content (r = 0.994) and therefore its values follow a similar pattern. Subira et al. [33] analyzed ancient and modern wheat species and concluded that with the development of agricultural technology, the amount of protein from wheat grains decreased. Protein decreased, leading to the increase of wheat carbohydrates, which are the most abundant fractions of wheat grains. According to our study, modern wheat grains have significantly higher values (*p* < 0.05) of carbohydrates and starch content compared to the ancient grains, which are in agreement with the results reported by others [6,20,34]. This may be explained by the enhanced grain yield of modern wheat being conducive to the “yield dilution phenomenon”, coupled with the higher ploidy level [6]. The physical characteristic of hectolitre weight is one of the wheat grain parameters which indicates milling potential [35]. The highest mean value of this parameter was obtained for common wheat, followed by einkorn and spelt. However, no significant (*p* < 0.05) differences were obtained for this parameter between species. The other physical characteristic, thousand-kernel mass, is one of the most important technological parameters which indicates grain quality [35]. According to our data, the highest value was obtained for spelt, followed by common wheat and einkorn. These findings are in agreement with those reported by Packa et al. [36], who also obtained higher TKM values for spelt than for common wheat. Einkorn had a significantly lower (*p* < 0.05) TKM value compared to spelt and common wheat. Similar data were reported by Bailliere et al. [4], who concluded that einkorn has a lower density than spelt and common wheat. It has been reported that TKM is well correlated with flour yield and therefore millers prefer kernels with higher TKM [24]. In this way, spelt seems to be more attractive. Contrarily, einkorn’s low TKM may affect its acceptance by millers.

The mineral elements of the wheat grain samples presented a large variation, which is in agreement with the data obtained by Krochmal-Marczak and Sawicka [37]. According to Simsek et al. [38], these variations may be due to the wheat variety age and a bran content reduction in relation to grain yield. However, from all the mineral elements analyzed, it has been reported that Ca changed very little during the historical period, being less associated with the change in varieties with improved yield. According to our data, the ancient wheat species presented high amounts of Ca compared to the modern varieties, which is in agreement with the data reported by others [39]. From all the analyzed species, spelt presented significantly high (*p* < 0.05) values of Ca compared to common wheat and einkorn. High values for mineral elements were also obtained for K and Cu in ancient species compared to the modern varieties. From all the analyzed mineral elements, Cu showed the lowest amount; these data are in agreement with that reported by Biel et al. [13]. The modern wheat varieties presented higher values for Fe and Na compared to the ancient species. This was unexpected, but may be due to a dilution effect that could occur over time. According to Fan et al. [40], a dilution effect has been present since 1968 until the present day, and decreasing trends of wheat mineral concentrations caused by increased grain yield and harvest index have been reported. Lower amounts of Fe for einkorn and spelt have also been reported by Zhao et al. [26]. From all the analyzed wheat species, the flours from einkorn grains presented the highest amount for Zn and Mn, whereas the flour from spelt grains had the highest amount of Mg. High levels of Mg and Ca for spelt have also been reported by Gomez-Becerra et al. [41]. Deficiencies in these two minerals may cause severe health problems, such as impaired bone growth, osteoporosis, inadequate bone mineralization and hypertension [19]. Therefore, spelt varieties with high amounts of Ca and Mg may be included in future breeding programs. However, all the analyzed wheat species showed low amounts of Fe and Zn. Different researchers have reported that wheat grains have significantly decreased in the content of these minerals since semi-dwarf cultivars were introduced in 1968 [40,42]. It has been concluded that this decrease is due to the partial dilution process, but also due to the fact that short-strawed varieties are less efficient at partitioning minerals to the kernel compared with the translocation of photosynthate. It seems that the mineral element content of wheat grains is strongly determined by genetics. The genetic differences in mineral content of the grains may be relevant to international efforts to improve health by increasing the wheat mineral content.

From the wheat structure, the pericarp (P), the seed coat (SC) and the aleurone layer (A) are the main components of the bran fraction, resulting from wheat milling. The SEM images of the wheat caryopsis characteristic shows that einkorn grains have a lower bran layer thickness than common wheat, indicating a higher proportion of endosperm in einkorn compared to common wheat. This is related to the smaller and lighter structure of einkorn seeds [28] in comparison with the common wheat seeds. As in other studies, it was observed that the cell walls of the starchy endosperm are very thin compared to the cell walls of the aleurone layer. Additionally, in all images of the starchy endosperm, starch granules of different shapes and sizes are highlighted and embedded in a protein matrix. The difference with the aleurone layer is that the protein appears rather as a continuous matrix, not as protein bodies. If we enlarge the images, the starchy endosperm appears with some voids (empty pockets), probably as the result of the loss of granules during the seed sectioning. In most studies, starch granules are classified into large, lens-shaped granules (type A) with a diameter of 15–40 µm and small, round-shaped granules (type B) with a diameter of 5–10 µm [28,43]. Additionally, type A granules appear first in the filling process of the wheat grains, and type B granules form later [44]. SEM analysis of wheat samples suggest that einkorn has more compact protein bonds compared to the common wheat and spelt grains, and the diameters of the type A starch granules in the protein matrix are smaller. According to the literature, the size of the starch granules has an influence on the starch digestibility, as small granules are easier to digest than the large ones [45]. Thus, we can affirm that einkorn, due to its small and smooth starch granules, may be easily attached to by hydrolytic enzymes, and is therefore one of the most bio-accessible and digestible types of cereals [27].

According to the wheat flours’ physicochemical characteristics, significant differences can be seen on the PCA graph between wheat species. Each wheat species has been clustered together. The ancient wheat species have been placed on the right part of the PCA graph and the modern ones on the left. This indicates the fact that these ancient species are not similar, being clearly distinguished by the PC1 axis. Moreover, einkorn species were placed on the bottom of the PCA graph, whereas spelt was placed on the upper right, indicating high similarities between their characteristics. The quality of ancient species was more related to their chemical characteristics: protein, gluten, lipid and ash content, all placed alongside PC2 axes on the right part of the PCA graph. These correlations are predictable, since ancient species have higher amounts of protein, gluten, lipid and ash content compared to modern species. Similar data have also been reported [20,29]. The high significant correlation (*p* < 0.05) between protein and gluten is explainable as gluten represents 75–85% of the total amount of wheat protein content [46]. Additionally, the high significant (*p* < 0.05) relationship between lipid and ash is because both are located in the germ and the wheat bran [20]. Due to the fact that ancient wheat species have a smaller size compared to the modern species, this will lead to wheat flours with higher bran content. Therefore, the ancient species will have more lipid and ash content and will be more closely associated with these chemical compounds on the PCA graph. Of all the wheat species, einkorn has the largest lipid and ash content, probably due to the fact that of all the analyzed species it has the smallest seeds. The germ represents a larger portion in small seeds, and for this reason the lipid content is higher in einkorn with respect to common wheat and spelt. The modern wheat varieties are more related to moisture, starch and carbohydrate content. The high significant correlation (*p* < 0.05) between starch and carbohydrate content is explainable, since starch is almost 75–80% of the carbohydrate content of the wheat grain [47]. There are high negative significant correlations between starch and protein (r = −0.920) distinguished by both PCA axes, due to the fact that during grain development, the processes of starch and protein synthesis are inversely correlated [48]. Additionally, test weight and TKM are more closely associated, with modern species being inversely correlated with protein and gluten content. Similar data have also been reported by Tremmel-Bede et al. [47]. In their study, grain Na, Mg and Fe were positively correlated (*p* < 0.05) with test weight and TKM. This fact indicates a correlation with kernel size, suggesting that plumper grain does not necessarily lead to smaller trace element concentration. This fact was unexplainable, since trace elements such as Fe are localized in the germ and aleurone layer of the wheat grains, which are in higher amounts in kernels of lower size. This finding is also in agreement with those of Zhao et al. [26], who also concluded that Fe amount is not dependent on grain kernel size. However, most of the analyzed mineral elements, Ca, K, Mn, Zn and Cu, were more associated with ancient wheat species, probably due to the fact that they are most located in the germ and the wheat bran, which exist in higher amounts in these grains.

## 4. Materials and Methods

### 4.1. Wheat Flour Samples

Twenty-four wheat varieties included in the active collection of the “Mihai Cristea” Vegetal Genetic Resources Bank (BRGV) of Suceava, Romania were analyzed in this study. Common wheat samples are represented by 5 Romanian varieties: Izvor, Glosa, Miranda, Andrada and Dumbrava; 3 Austrian varieties: Aurelius, Amicus and Tonnage from Austria; and 7 French varieties: Sofru, Sosthene, Sothys, Flavor, Solindo, Izalco and Sophie. Einkorn samples originate from Romania and include 1 local population (SVGB-11842) and 4 breeding lines (SVGB-11861, SVGB-11865, SVGB-11887, SVGB-11886). Spelt samples are represented by 2 Austrian varieties: Ebners Rotkorn and Frankenkorn; 1 Russian variety: Alkoran; and 1 German variety: Oberkulmer Rotkorn. All 24 samples were cultivated in September 2020 and harvested in July 2021, according to the agricultural period specific to winter wheat, on the experimental land of BRGV Suceava, in the same pedoclimatic conditions and without agrotechnical treatments being applied. The meteorological conditions during the study were obtained with the help of an automatic weather station belonging to the institution, which is located near the experimental field and which has the following GPS coordinates: 47°38′09″ N, 26°14′25″ E. The study was carried out in Suceava, a city in the NE region of Romania, which has a transitional temperate continental climate, and the local climate differences are due to the altitude and latitude, which determined that the average annual temperature dropped slightly from 10,41 °C in 2020 to 9.96 °C in 2021. Regarding the annual amount of precipitation, it was much lower in 2021 (276.9 mm) than in 2020 (493.9 mm). The climatic conditions corresponding to the wheat harvest period, namely July, indicate the driest month of the year, with the highest annual temperature (34.1 °C) and low precipitation (11 mm). After the harvesting stage, the seeds were kept in a cold room at a temperature of 3–5 °C until they were analyzed.

### 4.2. Physicochemical Composition of Wheat Flour Samples

The wheat samples were ground in a laboratory mill 3100 (Perten Instruments, Hägersten, Sweden), and the flours were then analyzed for the following physicochemical characteristics: moisture, ash, protein, wet gluten, lipids, starch, carbohydrates, test weight and thousand-kernel mass (TKM). The methods used for physicochemical composition of wheat flour samples are described below.

#### 4.2.1. Moisture Analysis

The moisture content was determined according to the standard method ICC 110/1 [49] and was expressed as a percentage of the weight of the original wheat sample. For this purpose, a quantity of 5 g of wheat flour was dried for 1.5 h at a temperature of 130 °C in a drying oven, then cooled in a desiccator and weighed.

#### 4.2.2. Ash Analysis

Ash content was determined according to ICC method 104/1 [50] by weighing 10 g of flour from each sample into crucibles and burning them in a muffle furnace at 900 °C until the cooled residue turned white. The ash content was expressed as a percentage of the weight of the residue relative to the original wheat sample.

#### 4.2.3. Protein Analysis

Protein content was determined via the Kjeldahl method (ICC method 105/2) [51] and using an automatic Kjeldahl analyzer (UDK127; VELP SCIENTIFICA, Milan, Italy), by measuring the total nitrogen compounds of wheat flour multiplied by the conventional factor of 5, 7. Thus, each flour sample (2 g), dried at 105 °C until it reached a constant weight, was accurately weighed and quantitatively transferred into a digestion tube. For digestion, the following reagents were added to each analyzed sample: 7 g K_2_SO_4_, 5 mg Se powder, 12 mL H_2_SO_4_ (96%) and 5 mL H_2_O_2_ (30%). The digestion tubes were heated for 20 min at 420 °C, then cooled to 50–60 °C and added to the distillation step in a digestion unit (UDK127; VELP SCIENTIFICA, Milan, Italy). The obtained distillates were collected in Erlenmeyer flasks, each containing 25 mL of boric acid (4%), to capture all the nitrogen, eliminating the risk of loss. Finally, the ammonia distilled from the samples was titrated with 0.2 N HCl, using bromocresol green as an indicator. The protein content, expressed as a percentage, was calculated using Equation (1):% Proteins = % Nitrogen × F(1)
where F is the conventional factor for wheat flour (5.7).

#### 4.2.4. Analysis of Wet Gluten Content

Wet gluten was determined according to ICC method 137/1 [52], using a gluten washer, Glutomatic 2200 (Perten Instruments, Hägersten, Sweden) with an 88 μm polyester sieve. A solution of 4.8 mL of salt was added to each 10 g of flour sample, and then mixed for 20 s to form the dough. After the mixing phase was finished, washing started automatically and continued for 5 min. Then, 30 s after the washing was finished, the wet gluten obtained was centrifuged for 1 min at 6000 rpm in a Gluten Index Centrifuge 2015 (Perten Instruments, Sweden) to remove the residual water adhering to the gluten, and then the total wet gluten weight was calculated. The wet gluten content was expressed as a percentage of the mass of the original sample (10 g).

#### 4.2.5. Analysis of Lipids

Following the ICC 136 method [53], fat content was determined via Soxhlet extraction using an automatic Soxhlet analyzer (SER 148/6; VELP, Milan, Italy). For this purpose, 5 g of flour sample was inserted into a porous cartridge to extract the fat with petroleum ether. Fat extraction in the Soxhlet analyzer was carried out by setting the work protocol as follows: immersion time of 60 min, removal time of 10 min, washing time of 60 min, recovery time of 20 min and cooling time of 5 min. After the analysis was finished, the extraction vessels were placed in a drying oven for 1 h at 105 °C, then cooled in a desiccator and weighed. The fat content was obtained from the difference between the initial weight of the sample and the weight of the dry residue after extraction. The results are expressed as a percentage (%) of total fat.

#### 4.2.6. Analysis of Total Starch

Total starch content was determined according to AACC approved method 76−13.01 [54] using a kit purchased from Megazyme International (Bray, Wicklow, Ireland) (100/100 tests per kit). Each of the wheat flour samples (~100 mg, accurately weighed), over which 0.2 mL of 80% (*v*/*v*) ethanol aqueous solution was added in order to moisten the samples and to help upon dispersal, were transferred to culture tubes. The tubes were agitated using a vortex mixer. Immediately, the samples were treated with sodium acetate buffer solution (100 mM, pH 5.0) and 3 mL each of thermostable α-amylase from Megazyme. The tubes were incubated in boiling water for 6 min, and then vigorously shaken after 2, 4 and 6 min to ensure complete homogeneity of the suspensions. Tubes containing 0.1 mL of amyloglucosidase (20 U) from the Megazyme kit were incubated in a water bath at 50 °C for 30 min. The transferred samples were diluted to 100 mL with distilled water and centrifuged for 10 min at 3000 rpm, and then the supernatant (clear, undiluted filtrate) was retained for analysis. To determine D-glucose, samples were treated with a glucose oxidase assay kit (GOPOD) from Megazyme and the absorbance was recorded at 340 nm on a UV-vis spectrophotometer (Shimadzu 3600, Tokyo, Japan).

#### 4.2.7. Analysis of Carbohydrates

Carbohydrate content was determined through differences by using the equation described by Alonso-Miravalles and O’Mahaony [55] (Equation (2)):% carbohydrates = 100 − (protein + fat + ash + moisture content)(2)

#### 4.2.8. Analysis of Test Weight and Thousand-Kernel Weight

Test weight was determined in accordance with the ISO 7971-1:2009 method [56]; thousand-kernel mass was determined using a seed counter (Contador, Pfeuffer, Kitzingen, Germany) and their mass was measured with an electronic balance (ISO 520:2010) [57].

### 4.3. Mineral Elements Analysis of Wheat Flour Samples

All the mineral composition analyses of samples were performed on an atomic absorption spectrometer (AAS) (AA-6300 SHIMADZU, Shimadzu Corporation, Kyoto, Japan) equipped with C_2_H_2_ flame excitation in the Instrumental Analysis Laboratory of the Faculty of Food Engineering, Suceava. The chemical analysis of the samples was carried out according to EN 14082:2003 [58].

The crucibles containing 10 g of each wheat sample were calcined in a Nabertherm LT 40/11/P330 calcination furnace (Nabertherm GmbH, Lilienthal, Germany), starting from an initial temperature, no higher than 100 °C, so that the temperature was raised by 50 °C/h until reaching 450 °C. Next, the samples were kept at 450 ± 25 °C overnight. Ash digestion was performed with concentrated hydrochloric acid (6 mol/L). The evaporation of the acid was carried out in the niche, on an electric plate, and the residue was dissolved with 10 mL of nitric acid (0.1 mol/L) (Sigma-Aldrich/Merck, Darmstadt, Germany). After digesting the ash samples, the containers were brought to the 50 mL mark using bidistilled and deionized water.

In order to obtain the calibration curves for each chemical element studied, stock solutions with the following concentrations were used: Ca—0.5, 1.0, 1.5, 3.0 and 5.0 mg/L; Mg—0.05, 0.1, 0.15, 0.2 and 0.3 mg/L; Zn—0.05, 0.15, 0.3, 0.45 and 0.6 mg/L; K—0.2, 0.4, 0.6, 0.8 and 1.0 mg/L; Na—0.1, 0.2, 0.3, 0.4 and 0.5 mg/L; Fe—0.5, 1.0, 1.5, 3.0 and 5.0 mg/L; Mn—0.5, 1.0, 1.5, 2.0 and 3.0 mg/L; Cu—0.5, 1.0, 1.5, 2.0 and 2.5 mg/L.

Each analysis method was set up to measure each sample five times, and then the mean of three samples was calculated. The values obtained from the AAS analyzes were used to calculate the real concentrations of elements using Equation (3):(3)$$\mathrm{Cmg}/\mathrm{kg}=C\;\mathrm{mg}/L\;\text{×}\;\mathrm{Dilution}\;\mathrm{factor}\;\text{×}\;\frac{V}{M}$$
where C is the real concentration of elements, V is the volume of the solution (50 mL) and M is the weight of the sample (10 g). The mineral elements were expressed as mg/kg on dry matter.

### 4.4. Microstructural Analysis

In order to study the microstructures of samples, they were dried in a Memmert oven (Memmert GmbH + Co. KG, Schwabach, Germany) at a temperature of 125 °C and then cross-sectioned. The morphology of the grains was determined using a scanning electron microscope (SEM), Vega II LMU model (Tescan, Brno-Kohoutovice, Czech Republic) equipped with an Everhart–Thornley secondary electron (SE) detector and backscattered electrons (BSE), which only works in a high-vacuum environment. The microstructure of the samples was studied in three points for each area and the most representative images were chosen. The acceleration voltage range was 30.00 kV [59], and the images were obtained at a magnification between 400–1000×. The images were analyzed with the help of ImageJ software (v. 1.45, National Institutes of Health, Bethesda, MD, USA) so that, after appropriate calibration, the diameters of starch granules were measured.

### 4.5. Statistical Analysis

The data obtained were statistically analyzed with the analysis of variance (ANOVA) method using the statistical package for social science (v.25, SPSS, Chicago, IL, USA). The principal component analysis was obtained using XLSTAT 2021.2.1 software (Addinsoft, New York, NY, USA). Tukey’s test was used at a significance level of *p* < 0.05.

## 5. Conclusions

There are considerable differences in the physicochemical composition and mineral elements between the ancient and modern species of wheat. Furthermore, the varieties of the wheat grains influenced their physicochemical composition. Generally, ancient wheat species presented higher values for chemical compounds of the grains, except for carbohydrate content. The moisture values were lower than 14% for all wheat species, indicating the long shelf life of the grains. The test weight did not present significant differences among the species, whereas thousand-kernel mass was significantly lower for einkorn, which may affect its acceptance by millers. The major mineral for all wheat species was K, followed by Mg and Ca. These macro elements were presented in the highest amount in spelt grains, which are recommended for consumption due to health reasons. Low levels were obtained for Fe and Zn for all the analyzed species, whereas Mn presented significantly higher values (*p* < 0.05) for einkorn grains. For Cu, modern grains presented significant lower values (*p* < 0.05) compared to ancient grains, whereas Na did not present high differences among species. The analysis of the microstructures of the analyzed wheat seeds highlighted the fact that einkorn grains contain less bran and, implicitly, more endosperm, than those of common wheat. Additionally, the compactness of the protein bonds and the smaller sizes of the starch granules make einkorn wheat a cereal with remarkable digestible properties. The PCA clearly shows differences among wheat species due to their physicochemical characteristics. This indicates that their composition is distinctive due to their genetic resource. Our data show that ancient wheat grains are a highly promising source of genetic diversity for protein. However, spelt seems to be the most promising genetic resource for mineral nutrients, particularly Ca and Mg. Our study is the first one to achieve a complete overview of the wheat grains’ physicochemical composition, including mineral element evaluation and microstructure analysis for different wheat varieties, which may be of great interest for wheat producers. This may be important data for Romanian agriculture, since in recent years there has been a growing demand for products with health benefits for consumers. Moreover, consumers also demand products with a high quality from a technological point of view. Taking into account the high protein and wet gluten content obtained for ancient species, this may lead to the conclusion that the bakery products obtained from them will be of a good quality for bread making. Moreover, the cultivated areas of ancient species are currently very low in Romania, but their high quality from a technological and nutritional point of view may lead to an increase in their cultivation area in order to obtain bakery products of a high quality which may satisfy current consumer demand.

## Figures and Tables

**Figure 1 plants-12-02138-f001:**
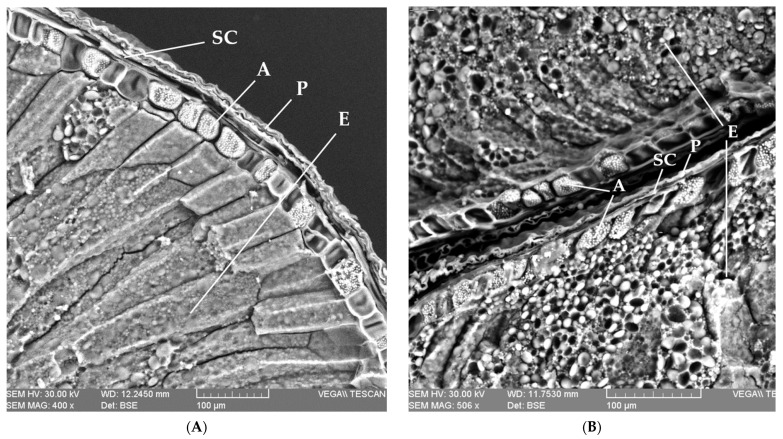
SEM micrographs of wheat caryopsis characteristic to the following species: common wheat (**A**) and einkorn (**B**). P—pericarp; SC—seed coat; A—aleurone layer; E—starchy endosperm.

**Figure 2 plants-12-02138-f002:**
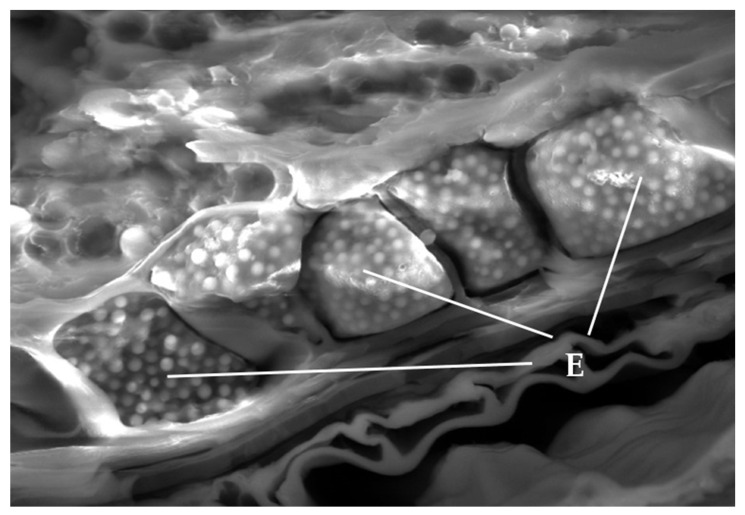
SEM micrograph of the aleurone layer (A) characteristic to einkorn: G—protein body globoids of the aleurone layer.

**Figure 3 plants-12-02138-f003:**
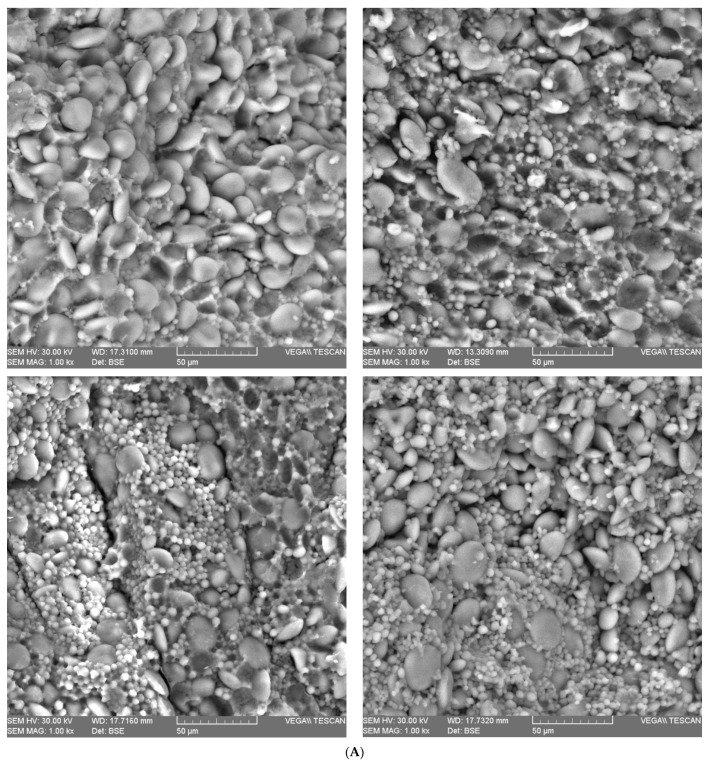
Scanning electron microscopy (SEM) micrographs of the endosperm of wheat samples: (**A**) common wheat; (**B**) einkorn; (**C**) spelt.

**Figure 4 plants-12-02138-f004:**
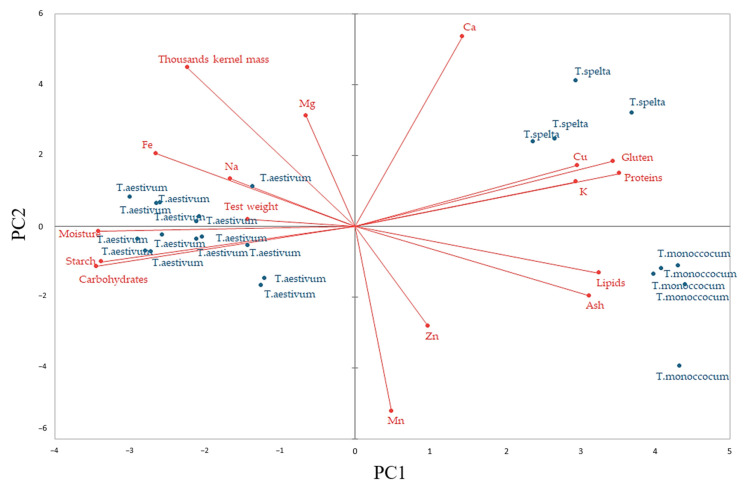
Principal component analysis between wheat flour characteristics.

**Table 1 plants-12-02138-t001:** Physicochemical characteristics of wheat samples.

Parameter	Wheat Species			F Value
	*Triticum aestivum*	*Triticum monococcum*	*Triticum spelta*	
Moisture (%)	13.24 ^b^ (12.48–13.76)	11.52 ^a^ (11.40–11.69)	11.78 ^a^ (11.68–11.85)	67.37 ***
Ash (%)	1.77 ^a^ (1.63–2.25)	2.47 ^b^ (2.34–2.61)	2.03 ^a^ (1.82–2.54)	26.07 ***
Protein (%)	12.83 ^a^ (10.52–14.43)	18.15 ^b^ (17.29–18.77)	18.95 ^b^ (18.24–19.27)	87.47 ***
Wet gluten (%)	27.8 ^a^ (21.00–33.00)	43.00 ^b^ (40.00–45.00)	47.50 ^b^ (45.00–49.00)	78.91 ***
Lipid (%)	1.77 ^a^ (1.42–2.27)	2.35 ^c^ (2.28–2.42)	2.07 ^b^ (1.91–2.31)	24.48 ***
Starch (%)	63.77 ^b^ (60.90–67.7)	56.18 ^a^ (55.90–56.4)	56.90 ^a^ (53.10–59.80)	34.48 ***
Carbohydrates (%)	70.44 ^b^ (67.96–73.31)	66.48 ^a^ (65.07–65.99)	65.15 ^a^ (64.03–65.91)	43.55 ***
Test weight (kg hl^−1^)	79.06 ^a^ (59–86)	73.68 ^a^ (71.2–75.7)	73.55 ^a^ (67.80–77.40)	2.42 ns
Thousand-kernel mass (g)	38.31 ^b^ (33.1–42.9)	24.92 ^a^ (19.60–28.1)	40.42 ^b^ (38.30–41.50)	45.90 ***

Values are expressed as mean and range. The minimum and maximum values of the wheat grains are indicated in brackets. ^a, b^—mean values in the same column followed by different letters are significantly different (*p* < 0.05). ns—*p* > 0.05, ***—*p* <0.0001.

**Table 2 plants-12-02138-t002:** Mineral composition of wheat samples.

Parameter	Wheat Species			F Value
	*Triticum aestivum*	*Triticum monococcum*	*Triticum spelta*	
Calcium (Ca) (mg kg^−1^)	232.87 ^a^ (189.03–277.78)	244.01 ^a^ (168.93–266.04)	428.46 ^b^ (380.69–469.12)	61.14 ***
Magnesium (Mg) (mg kg^−1^)	962.21 ^a^ (794.82–1147.09)	851.57 ^a^ (627.11–1142.51)	1016.37 ^a^ (771.24–1316.15)	1.38 ns
Potassium (K) (mg kg^−1^)	3504.28 ^a^ (2757.19–4347.18)	4782.30 ^a^ (3934.51–5404.34)	5029.32 ^a^ (4694.62–5609.11)	19.88 ***
Sodium (Na) (mg kg^−1^)	80.62 ^a^ (18.98–250.05)	25.35 ^a^ (19.82–33.21)	38.29 ^a^ (27.05–50.47)	2.19 ns
Zinc (Zn) (mg kg^−1^)	25.83 ^a^ (21.96–34.54)	30.36 ^a^ (17.07–34.13)	23.22 ^a^ (21.18–26.31)	3.55 *
Iron (Fe) (mg kg^−1^)	37.34 ^a^ (31.11–43.31)	21.90 ^a^ (19.93–23.13)	31.25 ^ab^ (23.63–41.58)	18.65 ***
Manganese (Mn) (mg kg^−1^)	58.54 ^ab^ (43.89–72.5)	74.45 ^b^ (52.78–93.36)	37.8 ^a^ (29.37–47.97)	13.46 ***
Copper (Cu) (mg kg^−1^)	2.30 ^a^ (0.82–3.36)	4.52 ^b^ (3.21–5.9)	4.25 ^b^ (3.07–5.42)	20.36 ***

All data were expressed on a dry-weight basis. Values are expressed as mean and range. The minimum and maximum values of the wheat grains are indicated in brackets. ^a, b^—mean values in the same column followed by different letters are significantly different (*p* < 0.05). ns—*p* > 0.05, *—*p* < 0.01, ***—*p* <0.0001.

## Data Availability

The datasets generated for this study are available on request to the corresponding author.

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
