# Peer review of "Physicochemical Characteristics and Microstructure of Ancient and Common Wheat Grains Cultivated in Romania"

_plants, 2023, doi:10.3390/plants12112138_

Round 1

Reviewer 1 Report

Manuscript title: Physicochemical characteristics and microstructure of ancient  and common wheat grains cultivated in Romania

Authors: Camelia Maria Golea, Silviu-Gabriel Stroe, Anca-Mihaela Gâtlan, Georgiana Gabriela Codină

 The manuscript on the presented topic and results is quite interesting not only for scientists but also for the grain and milling industry, however, I propose to introduce corrections to the manuscript based on the comments below.

 A certain drawback of the manuscript is that grain samples of 3 wheat species come only from 1-year studies and not at least 2-year studies.

 The abstract should be supplemented with the perspective of these studies, what specifically results from these studies for the society that consumes products derived from wheat grain

 Introduction is well written based on the available literature.

 Results:

Line 107-110, statement that... it is confirmed by the results of other studies or that it is consistent with the reports of the literature, please elaborate and describe it in detail on the basis of the cited literature,

Table 1, Table 2., please write what the letters a, b ... mean in the case of test results in table 1 and table 2, moreover, what do the given values mean, are they averages and ranges? e.g. Moisture (%) 13.24(12.48-13.76)b, it is better to put “b” next to the mean and superscript e.g. 13.24b (12.48-13.76).

Line 124-125, the statement (as in the case of Line 107-110) that ... by other authors, please elaborate and describe it in more detail on the basis of cited literature, this also applies to the rest of the Results chapter.

Table 2., full names of all mineral elements and their abbreviations should be given, e.g. Calcium (Ca), Magnesium (Mg), etc., then in Chapter 2.2. Wheat Flours Mineral Elements and throughout the manuscript, please use only elemental abbreviations, e.g. Ca, Mg, K, etc.

 Materials and Methods:

4.1. Wheat flour samples:

- I propose to list common wheat varieties, einkorn wheat varieties, spelled wheat varieties,

- provide GPS data for the field research site, were all 24 wheat varieties grown in the same place? Were these different growing locations for these varieties?

- briefly provide meteorological data during the growing season of wheat (temperature, precipitation),

- briefly describe the agrotechnical treatments (cultivation, fertilization, etc.) for the three wheat species tested,

- provide soil conditions,

4.2. Physicochemical composition of wheat flour samples:

- the test methods should be described in detail, assay methods and test apparatus as well as references to the literature should be provided

4.3. Mineral elements analysis of wheat flour samples and 4.4. Microstructural analysis

- provide references to the literature for the described research methods,

 Conclusions: So what are the prospects for growing einkorn and especially spelled wheat in Romania?

 References: should be carefully refined according to the recommendations of Plant

- for literature, e.g. 2, 3, etc., complete doi, this should be completed for all literature items

- for literature, e.g. 7, 11, etc., provide abbreviations of journals, this should be completed for all literature items

- for literature 32, 36, 37, translate the title into English

- for literature 44 (e.g. Grundas, S.T. Ultrastructure of the Grain, Flour, and Dough) and 47, 48 - provide full bibliographic data, all authors, year, doi, etc.

Author Response

15 May 2023

Dear Referee,

We would like to thank the referee for the close reading and for the proper suggestions. We hope that we provide all the answers to the reviewer’s comments.

Thank you very much for the recommendations to publish our paper entitled “Physicochemical characteristics and microstructure of ancient and common wheat grains cultivated in Romania”.

The present version of the paper has been revised according to the reviewer’s suggestions.             

We uploaded the corrected version of the article for which we used the red color for the addition text.

Reviewer comments

Manuscript title: Physicochemical characteristics and microstructure of ancient  and common wheat grains cultivated in Romania

Authors: Camelia Maria Golea, Silviu-Gabriel Stroe, Anca-Mihaela Gâtlan, Georgiana Gabriela Codină

The manuscript on the presented topic and results is quite interesting not only for scientists but also for the grain and milling industry, however, I propose to introduce corrections to the manuscript based on the comments below.

 A certain drawback of the manuscript is that grain samples of 3 wheat species come only from 1-year studies and not at least 2-year studies.

Response: We want to thank to the referee for the close reading of our manuscript. We corrected the manuscript according to the referee suggestions. Unfortunally, we do not have grain samples for 2-years study. However, our study is of a great interest since is the first study that makes a complete overview on the wheat grains physicochemical composition which includes mineral elements evaluation and their microstructure analysis for different wheat varieties cultivated in Romania (which is one of the most imporrtant producers in the EU and in the world). All the grains were cultivated by us in the same conditions due to the fact that we wanted to see properly the mineral content differences among wheat species. The largest part of the manuscript was dedicated to grain microstructure which is celarly different among wheat species. Also, we would like to thank the reviewer for all the comments and suggestions which have helped us to improve our paper.

Reviewer: The abstract should be supplemented with the perspective of these studies, what specifically results from these studies for the society that consumes products derived from wheat grain

Response: We want to thank to the referee for his/her suggestions. We completed our abstract with the perspective of our study in especially with the impact for our society that consumes products derived from wheat grain.

Reviewer:  Introduction is well written based on the available literature.

Response: We want to thank to the referee for his/her appreciations.

Reviewer:  Results: Line 107-110, statement that... it is confirmed by the results of other studies or that it is consistent with the reports of the literature, please elaborate and describe it in detail on the basis of the cited literature.

Response: We completed our statments with more describtions according to the referee suggestions.

Reviewer:  Table 1, Table 2., please write what the letters a, b ... mean in the case of test results in table 1 and table 2, moreover, what do the given values mean, are they averages and ranges? e.g. Moisture (%) 13.24(12.48-13.76)b, it is better to put “b” next to the mean and superscript e.g. 13.24b (12.48-13.76).

Response:We revised according to the referee suggestions.

Reviewer:  Line 124-125, the statement (as in the case of Line 107-110) that ... by other authors, please elaborate and describe it in more detail on the basis of cited literature, this also applies to the rest of the Results chapter.

Response: We completed our statments with more describtions according to the referee suggestions.

Reviewer:  Table 2., full names of all mineral elements and their abbreviations should be given, e.g. Calcium (Ca), Magnesium (Mg), etc., then in Chapter 2.2. Wheat Flours Mineral Elements and throughout the manuscript, please use only elemental abbreviations, e.g. Ca, Mg, K, etc.

Response:We revised according to the referee suggestions.

Reviewer:  Materials and Methods:

4.1. Wheat flour samples:

- I propose to list common wheat varieties, einkorn wheat varieties, spelled wheat varieties,

- provide GPS data for the field research site, were all 24 wheat varieties grown in the same place? Were these different growing locations for these varieties?

- briefly provide meteorological data during the growing season of wheat (temperature, precipitation),

- briefly describe the agrotechnical treatments (cultivation, fertilization, etc.) for the three wheat species tested,

- provide soil conditions,

Response: We completed the requested informations according to the referee suggestions.

Reviewer:  4.2. Physicochemical composition of wheat flour samples:

- the test methods should be described in detail, assay methods and test apparatus as well as references to the literature should be provided

Response: We completed in detail the methods used, the test apparatus and we provided the references according to the referee suggestions.

Reviewer:  4.3. Mineral elements analysis of wheat flour samples and 4.4. Microstructural analysis- provide references to the literature for the described research methods.

Response: We provided the references according to the referee suggestions.

Reviewer:  Conclusions: So what are the prospects for growing einkorn and especially spelled wheat in Romania?

Response: We completed the conclusions with some prospects for Romania.

References: should be carefully refined according to the recommendations of Plant

- for literature, e.g. 2, 3, etc., complete doi, this should be completed for all literature items

- for literature, e.g. 7, 11, etc., provide abbreviations of journals, this should be completed for all literature items

- for literature 32, 36, 37, translate the title into English

- for literature 44 (e.g. Grundas, S.T. Ultrastructure of the Grain, Flour, and Dough) and 47, 48 - provide full bibliographic data, all authors, year, doi, etc.

Response: We want to thank to the referee for the close reading of our manuscript. We revised the refrences according to the referee suggestions.

Sincerely,

Georgiana CODINÄ‚ et co.

Reviewer 2 Report

The article is correctly formatted, the discussion part is very comprehensive and informative. Photos of the endosperm and hulls of various wheat species are valuable. One note:

1. Please add figure 3 below that these are endosperm photos of wheat species.

Accept after this minor revision.

Author Response

15 May 2023

Dear Referee,

We would like to thank the referee for the close reading and for the proper suggestions. We hope that we provide all the answers to the reviewer’s comments.

Thank you very much for the recommendations to publish our paper entitled “Physicochemical characteristics and microstructure of ancient and common wheat grains cultivated in Romania”.

The present version of the paper has been revised according to the reviewer’s suggestions.             

We uploaded the corrected version of the article for which we used the red color for the addition text.

Reviewer comments

The article is correctly formatted, the discussion part is very comprehensive and informative. Photos of the endosperm and hulls of various wheat species are valuable. One note:

  1. Please add figure 3 below that these are endosperm photos of wheat species.

Accept after this minor revision.

Response: We want to thank to the referee for the close reading of our manuscript. We added the figure 3 below that these are endosperm photos of wheat species according to the referee suggestions.

Sincerely,

Georgiana CODINÄ‚ et co.

Reviewer 3 Report

This is a good manuscript and information is worth of publication.  Ancient wheat grains appear to have higher mineral content and less carbon hydrate, making it attractive to consumers.  If yield information is provided, it would be better for readers to understand if they are going to grow ancient wheat as an alternative wheat grain for niche market.

In the references, it is not consistent with the style, some are bolt for year but others are not.  Please change it based on the journal format requirement.

Other comments are marked on the margin of the manuscript.

English writing in general is pretty good except a few sentences that need clarification.

Author Response

15 May 2023

Dear Referee,

We would like to thank the referee for the close reading and for the proper suggestions. We hope that we provide all the answers to the reviewer’s comments.

Thank you very much for the recommendations to publish our paper entitled “Physicochemical characteristics and microstructure of ancient and common wheat grains cultivated in Romania”.

The present version of the paper has been revised according to the reviewer’s suggestions.             

We uploaded the corrected version of the article for which we used the red color for the addition text.

Reviewer comments

This is a good manuscript and information is worth of publication.  Ancient wheat grains appear to have higher mineral content and less carbon hydrate, making it attractive to consumers.  If yield information is provided, it would be better for readers to understand if they are going to grow ancient wheat as an alternative wheat grain for niche market.

Response: We want to thank to the referee for the close reading of our manuscript and the referee appreciations. We completed the manuscript with yield informations according to referee suggestions.

Reviewer comments: In the references, it is not consistent with the style, some are bolt for year but others are not.  Please change it based on the journal format requirement.

Response: We revised the refrences according to Plants style.

Reviewer comments: Other comments are marked on the margin of the manuscript.

Response: We completed the manuscript according to the referee suggestions.

Sincerely,

Georgiana CODINÄ‚ et co.

Reviewer 4 Report

English mother tongue proofreading

Uniform nomenclature of the three species in text and tables.

Report details of materials and methods (SR EN ISO 520, SR ISO 7971-1). Indicate the samples milling (mill type etc). Indicate the names of the genotypes studied for each cereal species. Indicate how many repetitions of the kernel’s microstructure were performed. Indicate if the mineral values ​​are expressed on dry matter and update the tables with this indication.

Standardize the term in the text and tables: test weight or hectoliter weight. Remove the comma and replace with point p<0.05, p<0.001, p<0.0001. Add that in brackets are indicated the minimum and maximum values.

Reduce from line 64 to 77. Line 42: only α-tocopherol is Vit. E; line 32: the range indicated for PC is too high for common wheat (7-18% dm.). line 69: only Einkorn has small size kernel (as can also be seen from the results reported in table 1). reduce from 72 to 83.

Table 2: mineral composition of wheat samples.

Table 3: after micrograph add : starch endosperm of..

Line 211-213: the humidity data is not consistent with indications of materials and methods (7%). Explain this difference.

Line 231: Explain what "lower and easier grain" means

Lines 232-244: condense.

246-247: rewrite the sentence, it is not clear.

Line 264: Explain what "with the launch year" means.

Line 273: Explain what the authors mean in the sentence "occur during the years"

329: I suggest indicating that the germ represents a larger portion in small seeds (einkorn type) than in wheat and for this reason the lipid content is higher in Einkorn with respect to common wheat.

English mother tongue proofreading

Author Response

15 May 2023

Dear Referee,

We would like to thank the referee for the close reading and for the proper suggestions. We hope that we provide all the answers to the reviewer’s comments.

Thank you very much for the recommendations to publish our paper entitled “Physicochemical characteristics and microstructure of ancient and common wheat grains cultivated in Romania”.

The present version of the paper has been revised according to the reviewer’s suggestions.             

We uploaded the corrected version of the article for which we used the red color for the addition text.

Reviewer comments: English mother tongue proofreading

Response: We want to thank to the referee for the close reading of our manuscript and the referee appreciations. All the manuscript have been now revised by an English teacher.

Reviewer comments: Uniform nomenclature of the three species in text and tables.

Response: We revised (uniformed) the nomenclature of the three species in text and tables according to the referee suggestions.

Reviewer comments: Report details of materials and methods (SR EN ISO 520, SR ISO 7971-1). Indicate the samples milling (mill type etc). Indicate the names of the genotypes studied for each cereal species. Indicate how many repetitions of the kernel’s microstructure were performed. Indicate if the mineral values ​​are expressed on dry matter and update the tables with this indication.

Response: We completed the manuscript with the requested informations according to the referee suggestions.

Reviewer comments: Standardize the term in the text and tables: test weight or hectoliter weight. Remove the comma and replace with point p<0.05, p<0.001, p<0.0001. Add that in brackets are indicated the minimum and maximum values.

Response: We want to thank to the referee for the close reading of our manuscript. We revised the manuscript according to the referee suggestions.

Reviewer comments: Reduce from line 64 to 77. Line 42: only α-tocopherol is Vit. E; line 32: the range indicated for PC is too high for common wheat (7-18% dm.). line 69: only Einkorn has small size kernel (as can also be seen from the results reported in table 1). reduce from 72 to 83.

Response: We revised according to the referee suggestions.

Reviewer comments: Table 2: mineral composition of wheat samples.

Response: We revised according to the referee suggestions.

Reviewer comments: Table 3: after micrograph add : starch endosperm of..

Response: We revised according to the referee suggestions.

Reviewer comments: Line 211-213: the humidity data is not consistent with indications of materials and methods (7%). Explain this difference.

Response: We really want to thank to the referee for the close reading of our manuscript. It was a mistake. The drying up to 7% is the procedure to keep the grains in the Bank Gene. This humidity was before the grains to be cultivated in the land. After the harvesting stage, the seeds were kept in a cold room at a temperature of 3-5ËšC until they were analyzed at the humidity of which were harvested. We revised in our manuscript.

Reviewer comments: Line 231: Explain what "lower and easier grain" means

Response: We rephrased the sentence. The explanations were from Azrani and Ashraf [6] which concluded: The endosperm accounts for approximately 90% of the dry weight of the grain. Compared to the ancient wheat, the lower protein content of modern wheat may be explained by its bigger and heavier grain that yields a larger starchy endosperm which, in turn, lessens its protein content.

Reviewer comments: Lines 232-244: condense.

Response: We condensed according to the referee suggestions.

Reviewer comments: 246-247: rewrite the sentence, it is not clear.

Response: We revised.

Reviewer comments: Line 264: Explain what "with the launch year" means.

Response: We rephrased. Generally, during the historic period the varieties have been improved. The launch year was the year when the same variety with an improved yield was launched. We rephrased to: ….. being less associated by the change to varieties with improved yield.

Reviewer comments: Line 273: Explain what the authors mean in the sentence "occur during the years"

Response: We completed the manuscript with explanations according to referee suggestions.

Reviewer comments: 329: I suggest indicating that the germ represents a larger portion in small seeds (einkorn type) than in wheat and for this reason the lipid content is higher in Einkorn with respect to common wheat.

Response: We completed the manuscript with explanations according to referee suggestions.

Sincerely,

Georgiana CODINÄ‚ et co.

Reviewer 5 Report

This manuscript presents an interesting study that comparing the grain physicochemical compositions, mineral elements levels and microstructures between current and ancient wheat cultivars. Data are well presented and the manuscript is carefully prepared. The results are clear and can support the conclusion nicely. Some English errors need to be corrected. e.g. line 418, should be K followed by Mg. 

Needs some revision.

Author Response

15 May 2023

Dear Referee,

We would like to thank the referee for the close reading and for the proper suggestions. We hope that we provide all the answers to the reviewer’s comments.

Thank you very much for the recommendations to publish our paper entitled “Physicochemical characteristics and microstructure of ancient and common wheat grains cultivated in Romania”.

The present version of the paper has been revised according to the reviewer’s suggestions.             

We uploaded the corrected version of the article for which we used the red color for the addition text.

Reviewer comments: This manuscript presents an interesting study that comparing the grain physicochemical compositions, mineral elements levels and microstructures between current and ancient wheat cultivars. Data are well presented and the manuscript is carefully prepared. The results are clear and can support the conclusion nicely. Some English errors need to be corrected. e.g. line 418, should be K followed by Mg. 

Response: We want to thank to the referee for the close reading of our manuscript and the referee appreciations. All the manuscript have been now revised by an English teacher. We corrected to the line 418 according to the referee suggestions.

Sincerely,

Georgiana CODINÄ‚ et co.

Round 2

Reviewer 1 Report

Dear Authors!

Thank you very much for the comprehensive and detailed revision of the manuscript.

Sincerely,

Author Response

Dear Referee,

Thank you very much for the appreciation.

Sincerely,

Codina et al.

Reviewer 4 Report

In the manuscript, I have noticed errors relating to the names of the species used in the experiment. I suggest to indicate at the first citation, in the Abstract and in the Introduction, the current name of each species and in brackets the Latin name, i.e. common Wheat (Triticum aestivum, L.), Spelt (Triticum spelta, L.) and Einkorn (Triticum monococcum L.). Authors must define a unique way to indicate the species, which could be, for brevity, the common name with the initial capital letter. 

Author Response

We want to thank to the referee for the close reading of our manuscript. We revised according to the referee suggestions